# Glycosaminoglycans influence regional mechanics in young but not old Achilles tendons

Jonathon L. Blank[1] , Jeremy D. Eekhoff[1] and Louis J. Soslowsky[1,2]

[1]*McKay Orthopaedic Research Laboratory, University of Pennsylvania, Philadelphia, Pennsylvania, USA*
[2]*Department of Bioengineering, University of Pennsylvania, Philadelphia, Pennsylvania, USA*

Handling Editors: Paul Greenhaff & Koyal Garg

The peer review history is available in the Supporting Information section of this article (https://doi.org/10.1113/JP286609#support-information-section).

**Abstract figure legend** Glycosaminoglycans (GAGs) are long, polysaccharide chains that are located between collagen fibrils in tendon and decrease with age in many musculoskeletal tissues. Yet, it is unknown whether declines in GAGs with age are responsible for age-related changes to Achilles tendon mechanics. The present study investigated Achilles tendon composition, structure and function following GAG depletion in young, middle-aged and aged mice. With age, cross-sectional area and the elastic modulus of the Achilles tendon insertion increased, and viscoelasticity of the Achilles tendon decreased. Yet, reductions in GAG content only influenced the elastic modulus of the Achilles tendon insertion in young mice, where the modulus was higher following GAG depletion. These findings suggest that GAGs may influence regional mechanical properties of the Achilles tendon.

**Jonathon Blank** completed his PhD in Mechanical Engineering at the University of Wisconsin-Madison (Madison, WI, USA) and is currently a Postdoctoral Fellow at the McKay Orthopaedic Research Laboratory at the University of Pennsylvania (Philadelphia, PA, USA). His research interests include non-invasive sensing of tendon structure–function and the role of extracellular matrix constituents in regulating tendon structure–function across age, following injury and during development of tendinopathy. Through this research he aims to enhance rehabilitation strategies and diagnosis of musculoskeletal disorders.

**Abstract**   Tendons are soft musculoskeletal tissues that transfer tensile loads from muscle to bone and consist of a highly organized extracellular matrix. Highly and repetitively loaded tendons such as the Achilles tendon are more susceptible to injury, and injuries are prevalent in the older population. Glycosaminoglycans (GAGs) are long polysaccharide chains that decrease in number with age in tendon and other musculoskeletal tissues. Yet, little is known of the role of GAGs in the tensile mechanics of the ageing Achilles tendon. The objective of this study was to investigate the mechanical role of GAGs in the ageing Achilles tendon. We enzymatically digested GAGs from the Achilles tendons of young, middle-aged and old C57BL/6 mice, removing a total of 64% of GAGs from the tendon insertion and midsubstance. GAG removal did not affect viscoelastic or structural properties across age in the Achilles tendon, paired with no changes to fibril realignment. However, removal of GAGs altered regional material properties at the Achilles tendon insertion in young mice in the absence of any material property changes to the Achilles tendon midsubstance. Finally, we found no changes to regional properties at the Achilles tendon insertion or midsubstance in middle-aged or old mice. In summary, GAG content influences regional mechanical properties at the calcaneal insertion of the Achilles tendon in young mice.

(Received 13 December 2024; accepted after revision 28 July 2025; first published online 25 August 2025)
**Corresponding author** L. J. Soslowsky: McKay Orthopaedic Research Laboratory, University of Pennsylvania, 307A Stemmler Hall, 3450 Hamilton Walk, Philadelphia, PA 19104-6081, USA.    Email: soslowsk@upenn.edu

## Key points

- The Achilles tendon is among the most commonly injured tendons, and aged tendon injuries are becoming an increasingly present societal burden.
- Glycosaminoglycans decrease across age in the Achilles tendon, yet their effect on structural properties, viscoelasticity and fibre realignment is minimal.
- In young Achilles tendons, glycosaminoglycans influence the elastic modulus near the calcaneal insertion, which was 60% greater following chondroitinase treatment.
- These data elucidate the mechanical role of glycosaminoglycans in the healthy Achilles tendon across age.

## Introduction

Injuries to soft tissues represent 45% of all 32 million musculoskeletal injuries per year (Butler et al., 2004), and tendons are among the most commonly injured tissues. Tendons transmit load from muscle to bone through a highly organized extracellular matrix (ECM), creating joint motion and enabling skeletal movement. Several factors influence the susceptibility of different tendon types to injury. For instance, many injuries occur in repetitive and high load-bearing tendons, such as the Achilles tendon (Järvinen et al., 2005; Leppilahti et al., 1996; Möller et al., 1996). Achilles tendon injuries in middle-aged and older adults are prevalent (Leppilahti et al., 1996; Möller et al., 1996) and are an increasing societal burden due to increased activity levels in older adults (Defroda et al., 2016; Huttunen et al., 2014). Following injury, tendons are slow to regain their native structure. This recovery process is poor in older adults (Leahy et al., 2022), resulting in permanent impairments

to tendon function and tasks of daily living. Further, altered ECM properties or composition could cause injury or impact recovery, and age-related changes to the tendon ECM may influence these factors.

The tendon ECM consists of a dense network of highly aligned collagen fibrils, fibres and fascicles organized in a hierarchy to resist tensile loads. While this tensile load-bearing structure is a common feature among different tendons, there are a variety of regional features that can accommodate the surrounding joint anatomy or specialized tendon functions. For example, the bone insertion of tendons consists of a highly specialized ECM with increased proteoglycan and fibrocartilage content (Waggett et al., 1998) and serves to minimize stress concentrations on the tendon attachment to bone (Thomopoulos et al., 2003, 2006). Further, the insertion of the Achilles tendon is subject to both compressive and tensile loads, depending on joint angle (Chimenti et al., 2016). In contrast, the tendon midsubstance is predominantly composed of highly aligned type 1 collagen,

which functions primarily to transmit tensile loading. While tendon injury (Huegel et al., 2015) and pathology (Saxena et al., 2013) can occur distinctly in one of these two regions, there is a gap in knowledge concerning how the mechanical properties of these regions are altered by advanced age in the Achilles tendon.

While collagen constituents (e.g. types I, III, V and XI) make up most of the tendon dry weight, many other small matrix proteins regulate important structure–function mechanisms. Glycosaminoglycans (GAGs) reside in the tendon ECM on proteoglycan core proteins such as decorin and biglycan (Yoon & Halper, 2005), which make up a small percentage of the tissue's dry weight (Eisner et al., 2022). As a negatively charged polysaccharide, GAGs can bind to water molecules to modulate tissue hydration, lubrication and friction between neighbouring structures in many musculoskeletal tissues, including tendon (Henninger et al., 2009; Rigozzi et al., 2013; Screen et al., 2005). While GAGs are present throughout tendon tissue, areas of high GAG concentration are often in regions of the tendon that undergo compression, such as the tendon insertion to bone (Waggett et al., 1998). However, the relative amount of GAGs in tendons with these compressed insertion regions declines with age (Riley et al., 1994).

The influence of GAGs on tensile tissue mechanics in tendon and similar tissues has been widely studied. Despite spanning interfibrillar space and connecting to adjacent fibrils (Cribb & Scott, 1995), GAGs do not act as mechanical crosslinkers since their digestion does not alter elastic or dynamic material properties in isolated tendon and ligament (Fessel & Snedeker, 2009; Lujan et al., 2007, 2009). Yet, an altered stress relaxation response in tendon fascicles suggests that GAGs may alter slower viscoelastic processes (Legerlotz et al., 2013). At the tendon insertion, GAGs are thought to modulate compressive properties (Aro et al., 2008) and may alter local tensile mechanics in this Achilles tendon region (Rigozzi et al., 2009). Yet, it is unknown whether this effect is age-specific and whether altered GAG content may give rise to inferior tissue mechanics with age in tendon. Therefore, the objective of this study was to investigate the mechanical role of GAGs in the ageing Achilles tendon. We hypothesized that GAGs would affect viscoelastic properties and regional mechanics near the Achilles tendon insertion, but that this effect would wane with age.

## Methods

### Ethical approval

All experiments were performed with approval from the University of Pennsylvania Institutional Animal Care and Use Committee (IACUC) under Protocol #806203.

The University of Pennsylvania IACUC is regulated by the Association for Assessment and Accreditation of Laboratory Animal Care International (AAALAC), the American Association for Laboratory Animal Science (AALAS) and the National Institutes of Health Public Health Service (PHS). The study conformed to the ethical principles and regulations outlined by *The Journal of Physiology* (Grundy, 2015).

### Tissue preparation

Male C57BL/6 (Charles River Laboratories, Wilmington, MA, USA) wild-type mice at postnatal days 150, 300 and 570 were used in this study ($n = 142$ mice total), corresponding to young, middle-aged and old adults. Animals were housed ($n = 5$ mice of equal age per cage) in a conventional facility with 12 h light/dark cycles and were fed standard chow and provided water *ad libitum*. The mice were euthanized by carbon dioxide inhalation, and were then weighed (P150: $30.6 \pm 3.0$ g, P300: $35.7 \pm 3.6$ g, P570: $38.4 \pm 6.0$ g). All animals were frozen at $-20°C$ until further use. Once thawed, Achilles tendons were harvested from mouse hindlimbs and cleaned from surrounding tissues with the calcaneus intact. The muscle was gently scraped from the proximal tendon. Each tendon was designated for one of two treatments (control or GAG digestion) and for either biomechanical testing, biochemistry or histology.

### Treatment

Immediately following fine dissection, each tendon–calcaneus unit was immersed in 0.33 mL of a Tris buffer supplemented with protease inhibitors (pH = 8.0) for 1 h at 37°C and under gentle agitation (Henninger et al., 2010). Following the initial buffer incubation, tendons were transferred to 0.33 mL of the same buffer solution (control), or the buffer solution supplemented with 0.5 U/mL chondroitinase ABC (cABC, Sigma #C3667, St Louis, MO, USA) (experimental treatment), which selectively cleaves sulphated GAGs from their proteoglycan core proteins. Tendons were incubated in the secondary buffer for 18 h at 37°C under gentle agitation. All tendons were rinsed in $1\times$ PBS following incubation. cABC concentration and incubation steps were chosen based on prior studies examining tendon and ligament mechanics following GAG depletion (Lujan et al., 2009; Rigozzi et al., 2009), and an 18 h incubation time was chosen to ensure high efficacy of the digestion using the 0.5 U/mL cABC concentration. All biochemical, biomechanical and histological procedures took place after this initial control or experimental treatment.

## Biochemistry

Biochemical assays were used to determine the composition of the whole tendon. GAG concentration was measured in each Achilles tendon using a 1,9-dimethylmethylene blue (DMMB) spectrophotometric assay ($n = 4$ per group) (Farndale et al., 1986). Following treatment, Achilles tendons were carefully dissected from their calcaneal insertion and lyophilized before recording the dry weight. Dry tendons were digested in a 5 mg/mL proteinase K–ammonium acetate solution overnight at 55°C. Following proteinase K digestion, the tendon solution was centrifuged at 10,000 $g$ for 10 min and 40 μL of the supernatant was transferred to a 96-well plate in triplicate. The DMMB solution was titrated to a pH of 1.5 to reduce the sensitivity of the reading to DNA and hyaluronic acid content (Zheng & Levenston, 2015). GAG concentration in the tendon was determined colorimetrically by reading absorbance at 525 and 595 nm using chondroitin sulphate as standard (0–32 μg/mL, linear fit $R^2 > 0.99$). All readings were normalized to the tissue's dry weight. Comparisons were made between contralateral limbs receiving either treatment.

Collagen content was measured following digestion using a hydroxyproline (OHP) assay to ensure that collagen was not degraded by incubation in the enzyme or buffer ($n = 4$–5 per group) (Stoilov et al., 2018). The tendon was hydrolysed with 12 N HCl in a sealed container at 110°C for 48 h. Samples were neutralized at 50°C in a fume hood for 48 h and resuspended in a citric acid buffer and 20 μL of the supernatant was transferred to a 96-well plate in triplicate. Collagen content was determined colorimetrically by reaction of the digest with 4-(dimethylamino)benzaldehyde and chloramine-T. All absorbance readings were taken at 550 nm using 1 mg/mL hydroxyproline as standard (0–6 μL, linear fit $R^2 > 0.99$). OHP was converted to total collagen content using a 1:14 ratio of OHP to collagen and normalized to the tissue's dry weight (Dourte et al., 2013). All OHP assays were performed on tendons from different mice that underwent biomechanical testing.

## Histology

Cryohistology was used to determine the regional composition of the Achilles tendon. Achilles tendons were harvested from frozen P150, P300 and P570 mouse hindlimbs ($n = 3$ per group) and treated according to the same treatment protocol. Preparation for imaging was designed based on a prior cryohistology protocol (Dyment et al., 2016). The Achilles tendon was fixed at 90° to the calcaneus in a cassette and immersed in 10% neutral-buffered formalin for 4 h. Samples were then soaked overnight in a 30% sucrose solution and embedded in optimal cutting temperature compound. Sections were cut sagittally using cryotape at an 8 μm thickness, glued to glass slides using 1% chitosan in acetic acid and dried overnight. Slides were then rehydrated in distilled water, stained in 0.025% toluidine blue for 3 min and rinsed, coverslipped using 30% fructose mounting medium, and imaged at 20× in a slide scanner (Zeiss Axioscan, Oberkochen, Germany). Following colour deconvolution, positive purple stain fraction in the insertion (0–1 mm from the insertion tidemark) and midsubstance (1–3 mm from the insertion tidemark) was used to quantify the presence of GAGs. Comparisons were made between contralateral limbs receiving either treatment.

## Biomechanics

The cross-sectional area (CSA) of the Achilles tendon insertion and midsubstance was recorded using a laser displacement sensor (2 μm resolution) (Favata, 2006). Four Verhoeff stain lines were applied to denote the insertion (0–1 mm from the calcaneal insertion), the midsubstance (1–3 mm from the calcaneal insertion) and the proximal grip location (4 mm from the calcaneal insertion). The calcaneus of the tendon was glued in rubber and secured in a custom grip, and the proximal tendon was glued in sandpaper and secured in a custom grip. The grip–tendon–grip structure was submerged in a 37°C 1× PBS bath (4 mm gauge length) on a material testing machine (Instron 5848) using a 10 N load cell. Specimens ($n = 20$ per group) were preloaded at 0.05 N and the grip-to-grip length was recorded. Specimens were preconditioned for 10 cycles from 0.75% to 1.25% axial strain at 0.25 Hz and allowed to recover for 60 s prior to viscoelastic testing. Two sequential stress relaxations were performed (1.5% and 3% strain, 600 s relaxation time), with each being followed by a frequency sweep from 0.1 to 10 Hz at an amplitude of 0.125% strain. Following viscoelastic testing, specimens were allowed to recover for 300 s. Specimens then underwent a quasi-static ramp-to-failure at 0.1% strain/s, during which optical strain and fibre realignment were recorded using quantified polarized light imaging (QPLI) in reflectance mode (FLIR Blackfly USBS, Newport, MA, USA) (Iannucci et al., 2023). Percentage relaxation, dynamic modulus and tan $\delta$ were recorded at each relaxation strain level. Stiffness, grip modulus and fibre realignment were recorded from the ramp to failure. A custom image registration script was used to track optical strain in the insertion and midsubstance regions (Fang & Lake, 2015) (MATLAB 2023a, Natick, MA, USA), where the slope of the linear region of the stress–strain curve in the insertion and midsubstance regions corresponded to the elastic modulus. Technical outliers (i.e. damaged during dissection or slipping from proximal grip) were excluded from analysis. All optical strain tracking was performed by a blinded investigator.

## Statistics

All quantitative data were assessed for statistical outliers (criterion: $>Q3 + 2.2 \times IQR$ or $<Q1 - 2.2 \times IQR$). We monitored all quantitative data for deviations from normality that would impact statistical results using Shapiro–Wilk tests. For OHP and biomechanics, the effects of age, digestion and their interaction were evaluated using two-way ANOVAs. Additionally, comparisons between the insertion and midsubstance modulus across age in control tendons were made using two-way repeated-measured ANOVAs. For QPLI measurements, the effect of strain, digestion and their interaction were evaluated within age groups using two-way repeated-measures ANOVAs. This same test for QPLI was also performed to make comparisons between age groups for control tendons. For histology and DMMB assays, matched contralateral limbs receiving either treatment were compared across age using two-way repeated-measures ANOVAs. Significant factors were

further investigated using *post hoc* paired *t* tests with Bonferroni's corrections. Significance was set at $\alpha = 0.05$.

## Results

### GAG depletion affects regional mechanics in young Achilles tendons

Histology showed that cABC treatment depleted GAGs similarly from the Achilles tendon midsubstance ($P = 0.0012$, Fig. 1*A* and *B*) and insertion ($P < 0.0001$, Fig 1*C* and *D*) across age. Additionally, there was heterotopic ossification (HO) in the Achilles tendon midsubstance of the P300 and P570 mice, as visualized by undigested areas of GAG concentration (red arrowheads, Fig. 1*A*). Generally, GAG content and collagen content decreased with age (Fig. 1*E* and *F*). When normalized to the average value for controls within each age group due to the presence of HO, our digestion protocol showed a

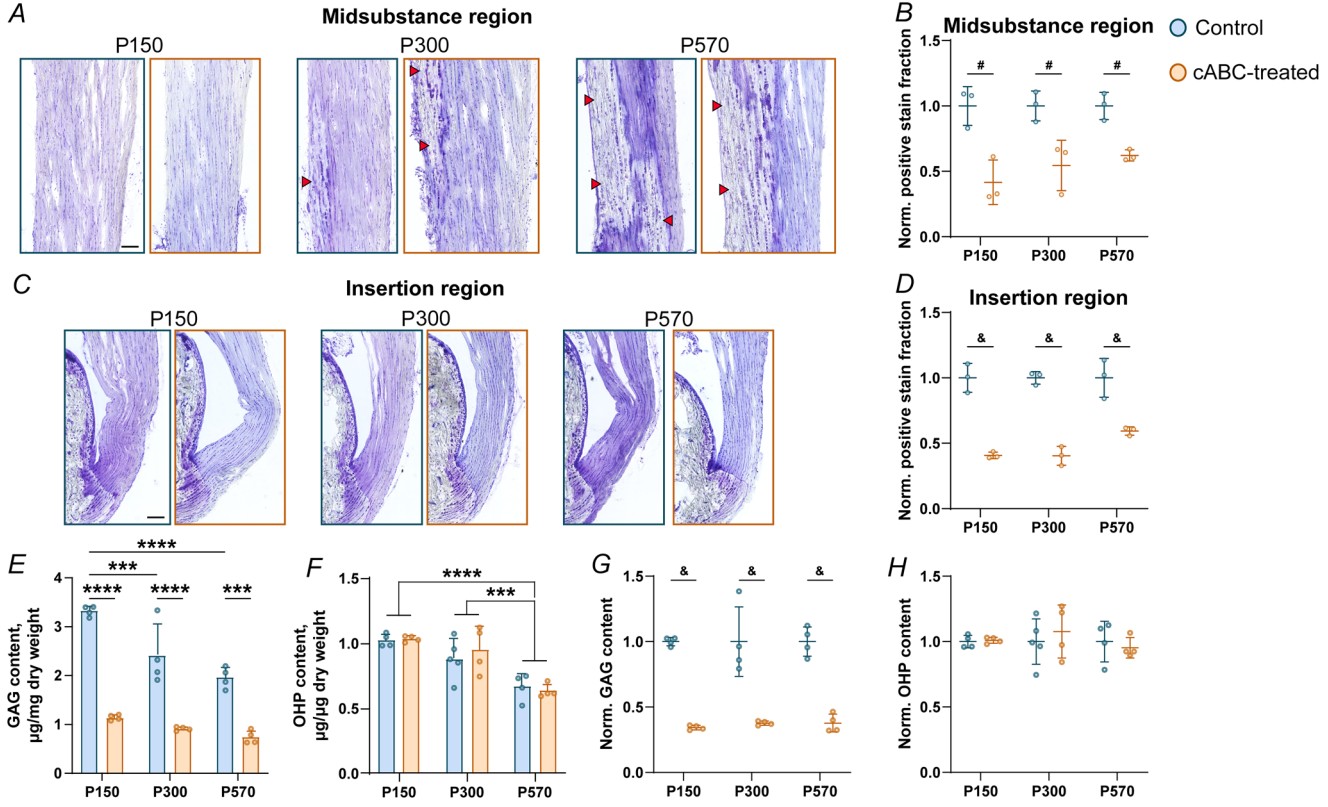

**Figure 1. GAG content across age and region following treatment in the Achilles tendon**
There were more GAGs in control tendons than in cABC-treated tendons in both the midsubstance (*A* and *B*) and insertion regions (*C* and *D*) (intensity range 0–200, scale bar = 100 μm). In the tendon midsubstance, there was a notable amount of heterotopic ossification with increased age, as shown by lack of consistent stain and mottled GAG concentrations (see red arrowheads). GAGs were depleted similarly across age in the midsubstance (*B*) and insertion (*D*) at P150 ($P = 0.0012$ and $P < 0.0001$, respectively). *E* and *F*, GAG and collagen content decreased similarly throughout age ($P = 0.0080$ and $P < 0.0001$, respectively). *G* and *H*, cABC treatment removed 64% of GAGs on average from the entire Achilles tendon ($P < 0.0001$) without any changes to collagen content ($P = 0.808$). Data are presented as mean ± SD (*$P < 0.05$, **$P < 0.01$, ***$P < 0.001$, ****$P < 0.0001$) (main effects, #$P < 0.01$, &$P < 0.0001$).

64% depletion of GAGs in the Achilles tendon across all ages ($P \leq 0.0001$) without changes to collagen content ($P = 0.808$) (Fig. 1*G* and *H*).

The overall Achilles tendon CSA increased with age ($P = 0.0033$), but there was no difference in CSA between the insertion and midsubstance region in any age group in control tendons ($P = 0.192$). Following digestion, there were no differences in CSA between the insertion and midsubstance regions of control and cABC-treated

tendons ($P = 0.660$ and $P = 0.714$, respectively), yet there were increases in CSA with age in both regions ($P = 0.0147$ and $P < 0.0001$, respectively) (Fig. 2*A* and *B*).

The elastic modulus of control tendons was higher in the midsubstance than the insertion at all ages ($P < 0.0001$), and the insertion modulus was greater in P570 control tendons than in P150 control tendons ($P = 0.0131$) (Fig. 2*E*). The elastic modulus at the Achilles tendon insertion was altered by cABC treatment, and

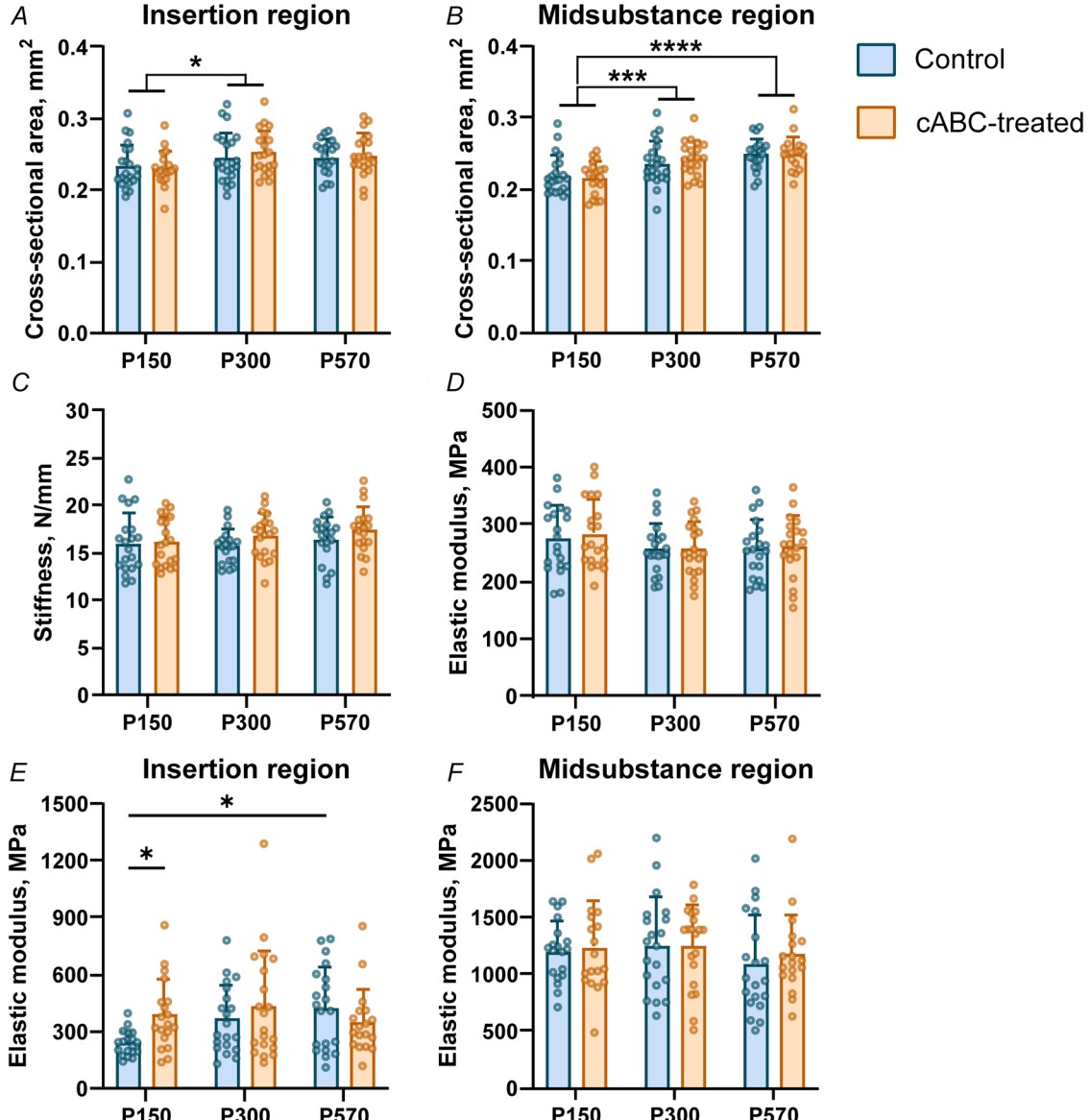

**Figure 2. Elastic biomechanics across age following treatment**
cABC treatment did not affect CSA in the Achilles tendon insertion (*A*) or midsubstance (*B*) regions, although CSA did increase with age in both regions ($P = 0.0147$ in the insertion, $P < 0.0001$ in the midsubstance). *C* and *D*, cABC treatment had no effect on tendon-level structural or material properties (stiffness and grip elastic modulus) in any age group. *E*, cABC treatment altered regional material properties at the Achilles tendon insertion ($P = 0.0495$), where there was an increase in elastic modulus in P150 mice but not in P300 or P570 mice. *F*, there was no change in regional material properties at the Achilles tendon midsubstance following cABC treatment. Data are presented as mean ± SD (*$P < 0.05$, ***$P < 0.001$, ****$P < 0.0001$).

the effect was age-specific ($P = 0.0495$). Specifically, the elastic modulus at the insertion region was greater in cABC-treated P150 tendons ($P = 0.0209$), but not in P300 or P570 tendons (Fig. 2$E$). cABC treatment had no effect on the elastic modulus of the Achilles tendon midsubstance ($P = 0.590$) (Fig. 2$F$).

### GAG depletion does not alter age-related changes in structural properties, viscoelasticity, or fibre realignment of the Achilles tendon

There were no changes to tissue-level stiffness ($P = 0.0952$) or elastic modulus ($P = 0.630$) in the Achilles tendons of any age group with GAG depletion (Fig. 2$C$ and $D$). There were changes between age groups in dynamic modulus ($P = 0.0005$ and $P = 0.0008$ for 1.5% and 3% axial strain, respectively) and tan $\delta$ ($P = 0.0027$ and $P = 0.0013$ for 1.5% and 3% axial strain, respectively) that are typical with advanced age (Fig. 3). However, cABC treatment had no effect on stress relaxation ($P = 0.488$ and $P = 0.888$ for 1.5% and 3% axial strain, respectively), dynamic modulus ($P = 0.915$ and $P = 0.929$ for 1.5% and 3% axial strain, respectively) or tan $\delta$ ($P = 0.395$ and $P = 0.384$ for 1.5% and 3% axial strain, respectively) at 1 Hz in the Achilles tendon. We did not detect changes in dynamic modulus or tan $\delta$ at 0.1, 5 or 10 Hz following cABC treatment.

Fibre realignment was not dependent upon age in the Achilles tendon insertion ($P = 0.730$) but was altered by age at high strain levels in the Achilles tendon midsubstance ($P = 0.0471$). However, cABC treatment had no effect on fibre realignment in the insertion or midsubstance region in P150 ($P = 0.276$ and $P = 0.572$, respectively, Fig. 4$A$ and $B$), P300 ($P = 0.528$ and $P = 0.772$, respectively, Fig. 4$C$ and $D$) or P570 ($P = 0.320$ and $P = 0.451$, respectively, Fig. 4$E$ and $F$) Achilles tendons.

## Discussion

The purpose of this study was to determine the role of GAGs in regulating tissue mechanics of the Achilles tendon across age. We found that the elastic modulus at the Achilles tendon insertion is much lower than at the midsubstance. Further, the Achilles tendon insertion elastic modulus increased following partial depletion of GAGs, and this effect was present in young tendons but not in middle-aged or old tendons. Finally, we found that GAGs have a negligible impact on structural properties, viscoelastic properties and fibril realignment across age.

Regional optical tracking of Achilles tendons showed a dramatic difference in modulus between the insertion and midsubstance regions of the tissue. Specifically, there was an increase in regional elastic modulus in the Achilles tendon midsubstance region by a factor of 2.5–5 as compared to the insertion region. Interestingly, the regional elastic modulus at the insertion increased by a factor of 1.6 when GAGs were digested from the tissue. Prior studies in the tendon and ligament midsubstance have found negligible impact of GAG digestion on mechanics (Fessel & Snedeker, 2009; Lujan et al., 2007, 2009). However, GAG content is generally higher in the tendon insertion than in the tendon midsubstance (Waggett et al., 1998), where there is increased compression, impingement and cartilaginous features as the tendon transitions to bone (Chimenti et al., 2016; Maffulli et al., 2006). The higher GAG content in this fibrocartilage-rich region of the Achilles tendon may modulate regional tissue elasticity, as has been found in isolated fibrocartilage (Han et al., 2016). Interestingly, a prior study examining regional tissue mechanics within the Achilles tendon midsubstance following GAG depletion found a decreased modulus in the distal portion of the Achilles tendon midsubstance in young tendons (Rigozzi et al., 2009), which contrasts with our finding of an increased modulus at the insertion following digestion. However, the distal region of tendon in the prior study was still proximal to the tendon insertion, whereas our regional measurements include the tendon insertion into bone and probably captures regions of the tendon near the insertion that have enriched GAG content. Finally, it is unknown whether GAGs would influence regional properties in humans, as there are differences between the mouse and human Achilles, including size, loading magnitudes relative to body weight and changes to the tendon hierarchy, where mouse Achilles tendons lack fascicles (i.e. bundles of fibres).

GAG content is known to decrease across age in many tissues, such as in cartilage (Riedler et al., 2017) and meniscus (Müller-Lutz et al., 2015). We show that GAG content relative to the dry weight of the ECM in the free Achilles tendon decreases from 0.33% in P150 tendons to 0.16% in P570 tendons, and similarly that collagen content in the free tendon decreased from nearly 100% collagen in P150 tendons to 66% collagen in P570 tendons. While decreases in GAG and collagen content have been found with age in adult tendon (Haut et al., 1992; Riley et al., 1994), the differences reported in this study are probably exaggerated due to increased heterotopic ossification in the rodent Achilles tendon across age (Fig. 1$A$) (Dai et al., 2020). Thus, decreases in GAG and OHP content with age in the mouse Achilles tendon are likely to be more modest than reported in this study. Concerning digestion, we found that decreases in positive GAG staining with digestion were similar across age in the Achilles tendon insertion and midsubstance. We also observed HO in the ageing tendon midsubstance, as visualized by undigested concentrations of GAGs in bony regions of the tendon (Fig. 1$A$), which may have contributed to less overall

GAGs being digested with increased age (i.e. 2.18 µg GAG per milligram tissue dry weight digested at P150 *vs.* 1.21 µg GAG per milligram tissue dry weight digested at P570). Interestingly, despite these regions of ossification in the ageing tendon midsubstance, we did not detect any differences in overall tendon stiffness, elastic modulus or midsubstance elastic modulus across age, indicating

that these HO deposits did not alter tissue-level tensile mechanics.

To our knowledge, there are no prior studies quantifying GAG content in the native human Achilles tendon. Riley et al. (1994) quantified GAG content in the human rotator cuff and biceps tendon, finding concentrations of 12.3 and 1.2 µg/mg, respectively.

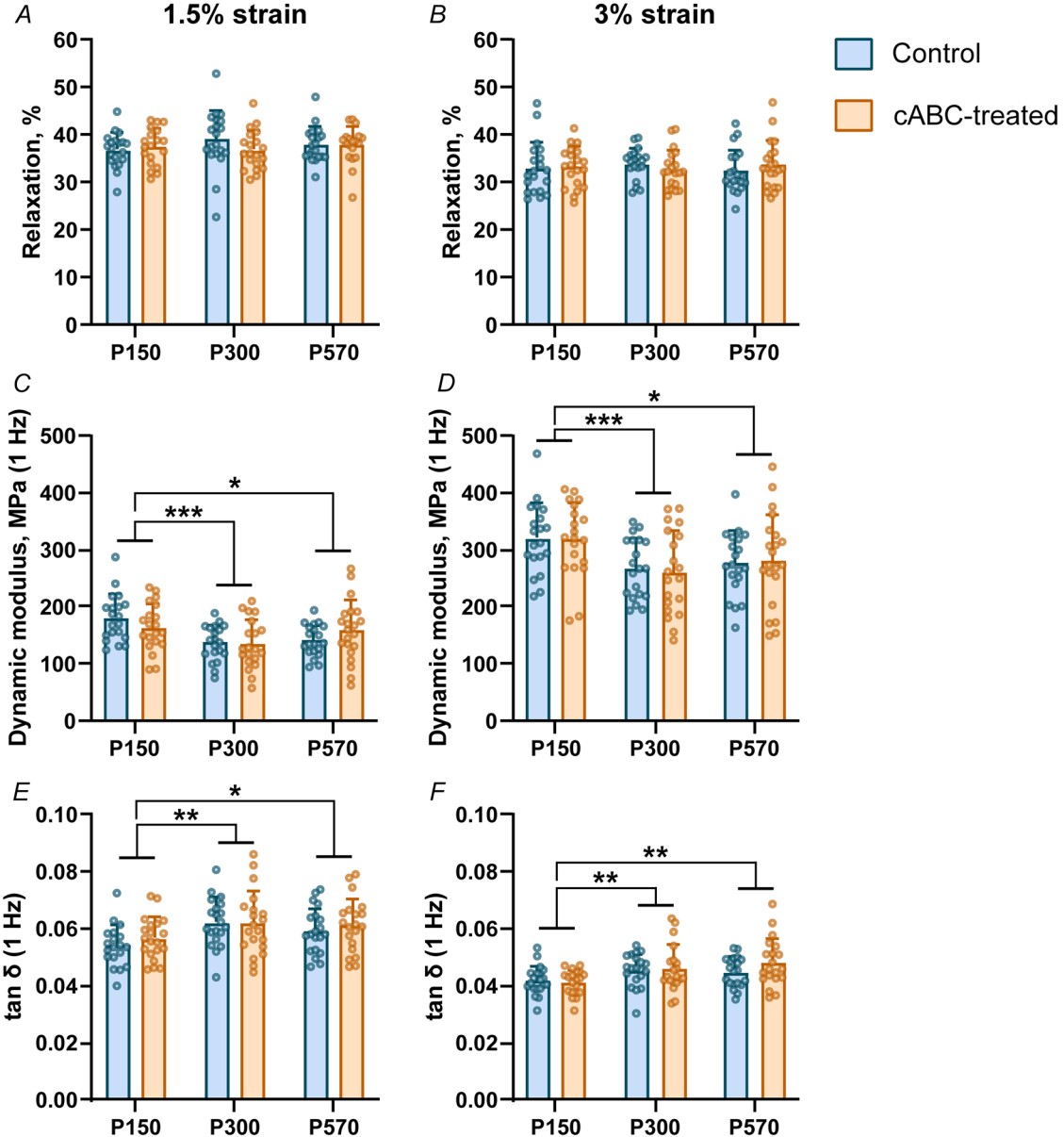

**Figure 3. Viscoelastic biomechanics across age following treatment.**
*A* and *B*, cABC treatment had no effect on stress relaxation in the Achilles tendon in any age group at either strain level. *C* and *D*, cABC treatment did not alter the dynamic modulus of the Achilles tendon in any age group at either strain level, although the dynamic modulus at P300 was lower than at P150 at both strain levels and the dynamic modulus at P570 was lower than at P150 at 3% strain. *E* and *F*, cABC treatment did not alter the phase angle (tan δ) of the Achilles tendon in any age group at either strain level, although tan δ was higher at P300 than at P150 at both strain levels and tan δ at P570 was higher than at P150 at 3% strain. Similarly, no changes in viscoelasticity at 0.1, 5 or 10 Hz were found following treatment. Data are presented as mean ± SD (*$P < 0.05$, **$P < 0.01$, ***$P < 0.001$).

Our measured GAG content in the healthy mouse Achilles tendon (3.31 µg/mg at P150, 2.40 µg/mg at P300 and 1.95 µg/mg at P570) is between the range of these concentrations, yet there is considerable variability in GAG content between tissue types. Changes in GAG content with age may also be tissue- and environment-dependent, as a prior study in murine flexor digitorum longus tendons found a similar decrease in GAG content with age when normalized to the dry weight (Mlawer et al., 2024), and increased GAG synthesis in younger tendons when stress-deprived (Connizzo et al., 2020). Investigating whether the Achilles tendon can modulate GAG production to accommodate changes in loading environment may elucidate age-related

pathogenic changes to tendon associated with altered GAG content, such as in insertional Achilles tendinopathy (Bah et al., 2020; Chimenti et al., 2017).

Changes in GAG content across age do not explain age-related changes in tensile tissue mechanics of the Achilles tendon. We found that cABC treatment had no effect on tissue-level structural properties, viscoelastic properties or fibril realignment in any age group. More specifically, while we corroborated typical findings of altered CSA and viscoelasticity in rodent tendons with age (Figs 2 and 3) (Dunkman et al., 2013, 2014; Pardes et al., 2017), cABC treatment did not alter these measurements within any age groups. This finding corresponds with prior studies in tendon (Fessel & Snedeker, 2009) and

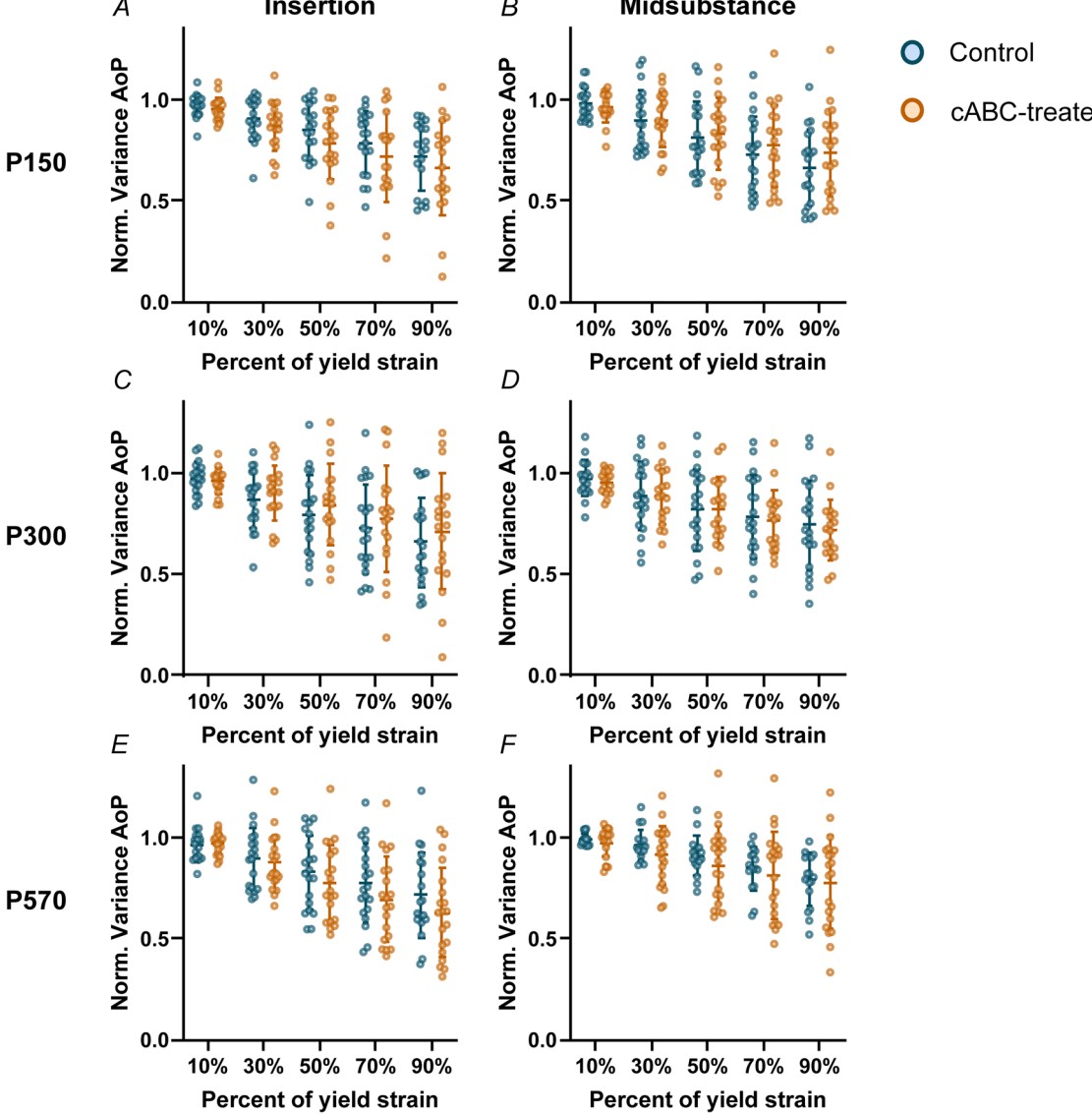

**Figure 4. Fibril realignment patterns across age and treatment**
We did not observe changes in fibril realignment patterns (angle of polarization, AoP) in either the insertion (*A*, *C* and *E*) or midsubstance (*B*, *D* and *F*) regions in P150 (*A* and *B*), P300 (*C* and *D*) or P570 (*E* and *F*) Achilles tendons. Data are shown as mean ± SD.

ligament (Lujan et al., 2007, 2009) that demonstrated no effects of GAG digestion on tensile stiffness or dynamic modulus, and that fibril realignment patterns are not solely descriptive of material property changes at the insertion (Connizzo et al., 2013). Atomic force microscopy and related microscopy techniques have found that GAGs alter shear deformation (Muljadi & Andarawis-Puri, 2023) and promote spacing (Screen et al., 2006) and sliding along fibrils (Al Makhzoomi et al., 2022; Rigozzi et al., 2013), possibly due to fibril lubrication. Whether these microscale mechanisms for GAGs occur in the Achilles tendon and explain age-related changes to stiffness or viscoelasticity are an area of future study.

There are a few limitations to consider when interpreting the results of this study. First, we chose to examine male mice because middle-aged men are at higher risk of Achilles tendon ruptures (Maffulli et al., 1999). However, GAG content and synthesis in tendon may be sex-specific (Connizzo et al., 2020; Kobayashi-Miura et al., 2022), and future studies should consider sex as a biological variable that could alter GAG content or the interaction of GAGs with the tissue microenvironment. Second, we chose to test tendons with the Achilles tendon at an angle of 180° relative to the calcaneus to maximize the available in-plane insertion region for optical tracking, which included the first millimetre of tissue up to the insertion into bone. In rodents, a more neutral ankle angle may better reflect tensile mechanics in the Achilles tendon *in situ* (Kurtaliaj et al., 2019). Third, a higher efficacy digestion protocol (i.e. greater than 64%) such as incubation for a longer time, using a secondary digestion (Schmidt et al., 1990) or using an increased cABC concentration (i.e. 1 U/mL) may have increased the mechanical changes observed with GAG removal. Finally, performing region-specific biochemical assays would have provided additional explanatory power for regional differences in the mechanical changes following GAG removal.

There are many applications of these findings to orthopaedic conditions where an excess of GAGs are present. There are notably more GAGs present following tendon rupture and onset of tendinopathy (Choi et al., 2016; Fu et al., 2007). These tendons have remarkably different mechanics, most notably a decreased stiffness in humans with Achilles tendinopathy (Arya & Kulig, 2010). Metabolic diseases such as mucopolysaccharidoses (MPS) are characterized by a lack of production of enzymes to break down GAGs (Khan et al., 2017), and recent studies have shown decreases in tendon modulus in canine Achilles tendons with MPS I and an excess of GAGs (Lau et al., 2024). Understanding the mechanical and biological role of GAG deposits in these conditions will be useful in developing new therapies or treatment interventions.

This study helps to elucidate the role of GAGs in tensile tendon mechanics throughout ageing, finding that GAGs may influence regional properties at the Achilles tendon insertion in young tendons. Further, age-related changes in GAG content are not responsible for changes to structural properties, viscoelasticity or fibril realignment patterns. Future work should consider the effect of GAG accumulations on Achilles tendon structure and function.

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

## Additional information

### Data availability statement

The data supporting the findings in this study are partially available in the supporting information. All data are available from the corresponding author upon reasonable request.

### Competing interests

The authors declare no conflicts of interest in relation to the data presented.

### Author contributions

J.B., J.E. and L.S. designed the experiments and wrote the manuscript. J.B. and J.E. treated specimens and developed standard operating procedures. J.E. blinded all investigators. J.B. collected tissues and performed biomechanics, biochemistry, histology and all post test analyses, and prepared the first manuscript draft. All authors have approved of the final version of the manuscript. All persons designated as authors qualify for authorship.

## Funding

NIH/NIAMS F32AR082671, Penn Center for Musculoskeletal Disorders (NIH/NIAMS P30AR069619).

## Acknowledgements

The authors would like to acknowledge Stephanie Weiss (Soslowsky Lab, UPenn) for her assistance with the mouse colony.

## Keywords

Achilles tendon, ageing, extracellular matrix, foot and ankle, imaging

## Supporting information

Additional supporting information can be found online in the Supporting Information section at the end of the HTML view of the article. Supporting information files available:

**Peer Review History**
**Supplementary Tables**

