## [Peer Review History · The Journal of Physiology]

Glycosaminoglycans influence regional mechanics in young but not old Achilles tendons

Jonathon L Blank, Jeremy Eekhoff, and Louis Soslowsky

DOI: 10.1113/JP286609

Corresponding author(s): Jonathon Blank (jonathon.blank@pennmedicine.upenn.edu)

Review Timeline:

Submission Date:	13-Dec-2024
Editorial Decision:	14-Feb-2025
Revision Received:	04-Mar-2025
Editorial Decision:	04-Apr-2025
Revision Received:	12-Jun-2025
Accepted:	28-Jul-2025

Senior Editor: Paul Greenhaff

Reviewing Editor: Koyal Garg

Transaction Report:

Dear Dr Blank,

Re: JP-RP-2024-286609 "Glycosaminoglycans influence regional mechanics in young but not old Achilles tendons" by Jonathon Blank, Jeremy Eekhoff, and Louis Soslowsky

Thank you for submitting your manuscript to The Journal of Physiology. It has been assessed by a Reviewing Editor and by 2 expert referees and we are pleased to tell you that it is potentially acceptable for publication following satisfactory major revision.

Please address all the points raised and incorporate all requested revisions or explain in your Response to Referees why a change has not been made. We hope you will find the comments helpful and that you will be able to return your revised manuscript within 2 months. If you require longer than this, please contact journal staff: jp@physoc.org. Please note that this letter does not constitute a guarantee for acceptance of your revised manuscript.

REVISION CHECKLIST:

Please upload two versions of your manuscript text: one with all relevant changes highlighted and one clean version with no

changes tracked. The manuscript file should include all tables and figure legends, but each figure/graph should be uploaded as separate, high-resolution files.

We look forward to receiving your revised submission.

Yours sincerely,

Paul Greenhaff
Senior Editor
The Journal of Physiology

REQUIRED ITEMS FOR REVISION

- Author photo and profile. First or joint first authors are asked to provide a short biography (no more than 100 words for one author or 150 words in total for joint first authors) and a portrait photograph. These should be uploaded and clearly labelled together in a Word document with the revised version of the manuscript. See Information for Authors for further details.

- You must start the Methods section with a paragraph headed Ethical approval (https://jp.msubmit.net/cgi-bin/main.plex?form_type=display_requirements#methods).

Research must comply with The Journal's policies regarding animal experiments (<https://physoc.onlinelibrary.wiley.com/hub/animal-experiments>) and adherence to these policies must be stated in the manuscript.

Authors should confirm in their Methods section that their experiments were carried out according to the guidelines laid down by their institution's animal welfare committee, including an ethics approval reference number. The Methods section must contain a statement about access to food, water and housing, details of the anaesthetic regime: anaesthetic used, dose and route of administration, and method of killing the experimental animals.

- The reference list must be in alphabetical order, rather than numbered, to comply with our Journal format.

- Your manuscript must include a complete Additional Information section, including competing interests; funding; author contributions and acknowledgements.

- Please upload separate high-quality figure files via the submission form.

- You must upload original, uncropped western blot/gel images (including controls) if they are not included in the manuscript. This is to confirm that no inappropriate, unethical or misleading image manipulation has occurred. These should be uploaded as 'Supporting information for review process only'. Please label/highlight the original gels so that we can clearly see which sections/lanes have been used in the manuscript figures. For more information, see: <https://physoc.onlinelibrary.wiley.com/hub/journal-policies#imagmanip>.

- Your paper contains Supporting Information of a type that we no longer publish, including supplementary tables and figures. Any information essential to an understanding of the paper must be included as part of the main manuscript and figures. The only Supporting Information that we publish are video and audio, 3D structures, program codes and large data

files. Your revised paper will be returned to you if it does not adhere to our Supporting Information Guidelines.

- Papers must comply with the Statistics Policy: https://jp.msubmit.net/cgi-bin/main.plex?form_type=display_requirements#statistics.

In summary:

- If n {less than or equal to} 30, all data points must be plotted in the figure in a way that reveals their range and distribution. A bar graph with data points overlaid, a box and whisker plot or a violin plot (preferably with data points included) are acceptable formats.
- If $n > 30$, then the entire raw dataset must be made available either as supporting information, or hosted on a not-for-profit repository, e.g. FigShare, with access details provided in the manuscript.
- 'n' clearly defined (e.g. x cells from y slices in z animals) in the Methods. Authors should be mindful of pseudoreplication.
- All relevant 'n' values must be clearly stated in the main text, figures and tables.
- The most appropriate summary statistic (e.g. mean or median and standard deviation) must be used. Standard Error of the Mean (SEM) alone is not permitted.
- Exact p values must be stated. Authors must not use 'greater than' or 'less than'. Exact p values must be stated to three significant figures even when 'no statistical significance' is claimed.

- A Data Availability Statement is required for all papers reporting original data. This must be in the Additional Information section of the manuscript itself. It must have the paragraph heading 'Data Availability Statement'. All data supporting the results in the paper must be either: in the paper itself; uploaded as Supporting Information for Online Publication; or archived in an appropriate public repository. The statement needs to describe the availability or the absence of shared data. Authors must include in their statement: a link to the repository they have used, or a statement that it is available as Supporting Information; reference the data in the appropriate sections(s) of their manuscript; and cite the data they have shared in the References section. Whenever possible, the scripts and other artefacts used to generate the analyses presented in the paper should also be publicly archived. If sharing data compromises ethical standards or legal requirements then authors are not expected to share it, but must note this in their statement. For more information, see our Statistics Policy.

- Please include an Abstract Figure file, as well as the Figure Legend text within the main article file. The Abstract Figure is a piece of artwork designed to give readers an immediate understanding of the research and should summarise the main conclusions. If possible, the image should be easily 'readable' from left to right or top to bottom. It should show the physiological relevance of the manuscript so readers can assess the importance and content of its findings. Abstract Figures should not merely recapitulate other figures in the manuscript. Please try to keep the diagram as simple as possible and without superfluous information that may distract from the main conclusion(s). Abstract Figures must be provided by authors no later than the revised manuscript stage and should be uploaded as a separate file during online submission labelled as File Type 'Abstract Figure'. Please also ensure that you include the figure legend in the main article file. All Abstract Figures should be created using BioRender. Authors should use The Journal's premium BioRender account to export high-resolution images. Details on how to use and access the premium account are included as part of this email.

- Please include a full title page as part of your main article (Word) file, which should contain the following: title, authors, affiliations, corresponding author name and contact details, keywords, and running title.

EDITOR COMMENTS

This is an interesting and well written/executed study regarding a controversial topic in the field. Addressing the reviewer comments will strengthen the manuscript for publication.

Senior Editor:

This manuscript has been considered by a reviewing editor and two expert reviewers. The study is well powered and all believe this original research is well written. The Reviewing Editor and Reviewer 2 show enthusiasm (the latter has raised several points that the authors need to address). However, Reviewer 2 is less enthusiastic about the scientific impact of the work described and has raised a number of major concerns. These include a need for regional tendon analyses, and that all mice (not only the young) should have been included in histological analysis. It is important the authors address these concerns in a comprehensive manner when revising the manuscript, including the inclusion of additional new data if possible. Thank you for considering the Journal of Physiology to publish this work and we look forward to receiving the revised manuscript.

REFEREE COMMENTS

Referee #1:

This manuscript examines the role of GAGs on Achilles tendon mechanical properties during aging. The premise of the manuscript is well presented in that tendon mechanical properties changes and the GAG content also changes with age so do GAGs play a role in the mechanical changes. The manuscript is well written. Several concerns with this manuscript are the lack of full analysis of the GAGs biochemically (full tendon analysis instead of regional analyses given the expected GAG difference between insertion and midsubstance; only the young mice analyzed histologically) and the limited new information generated compared to previous literature.

Specific Comments

Line 128: If it is known that the insertion and midsubstance have different biochemical compositions, then why were they not examined separately?

Line 152: Why were the middle and old age tendons not examined?

Lines 224 - 228: Is the difference in the treated tendons different than the increases seen with the control samples for age?

Figure 1: It seems very odd that the reduction is the exact same across all 3 ages when they older tendons have less GAGs. Please provide an explanation of this finding.

Figure 2: It seems panel D is mislabeled and should be stiffness of the Midsubstance region. Please clarify how the midsubstance stiffness remains unchanged and the midsubstance CSA significantly changes with age, but there is no effect on the elastic modulus.

Referee #2:

This study evaluated the role of GAGs in tensile mechanics of tendons from young, middle-aged, and old adult mice. This topic has been debated for many years, with previous studies sometimes drawing contradictory conclusions about the role of GAGs in dictating tendon mechanical properties. The current study was well organized, executed with strong experimental techniques/analysis, and summarized clearly in a well-written manuscript. There are a few questions and concerns that should be addressed:

- How was the incubation time selected? Does the GAG removal reach a plateau or would a longer incubation time cause further reduction in GAGs? Do other tissue components start to break down (e.g., collagen) under longer enzymatic incubation times? Also, were tendons removed from the chondroitinase ABC and rinsed at the conclusion of the 18-hour period or kept in the solution until testing?
- The histological images shown for P150 tendons in Figure 1A and 1B are beautiful and insightful. Why was histological analysis not performed for the P300 and P570 mice? It would be very helpful to include images from these two other ages so that similarities and differences by age can be observed. In addition, the positive stain fraction values could be

computed/reported for the older mice too.

- What is the hypothesized mechanism for why insertion elastic modulus increased in P150 tendons following GAG depletion?
- Figure 4 only shows data for P150 (a-b) but the caption suggests that P300 (c-d) and P570 (e-f) were meant to be included. Where are these additional data?
- If the collagen content (i.e., OHP) and GAG content both decrease in the P570 tendons, what compositional constituents of these older tendons are increased? In other words, what are these tendons becoming with increased age?
- Can the authors discuss how GAG composition in mouse Achilles tendons compares to human Achilles tendons? Are the relative percentages of GAG similar or quite different across the two species? And what is known about GAG degradation with age in human Achilles tendons? What implications does this have for interpreting the current results in the context of human tissues?
- The assertions that GAGs (1) "likely regulate the elastic modulus" in the insertion region and (2) "modulate regional properties at the Achilles tendon insertion in young tendons" seem too strong given that cABC effects were only (and just barely) significant in young tendons ($p=0.0495$). Many of the data points for the ChABC-treated P150 tendons fell within the range of untreated tendons (Figure 2E), and it appears that larger modulus values from four tendons (~600-900 MPa) are likely driving the statistical difference between untreated and treated groups. That's not to say these aren't groups aren't different...but just that conclusions based on these differences should be appropriately tempered.
- In the discussion, is the sentence that reads: "While decreases in GAG and collagen content have been found with age in adult tendon [19],[39] ..." referring to adult human tendon? Also, does the sentence: "Thus, decreases in GAG and OHP content with age in the Achilles tendon are likely to be more modest than reported in this study" refer to human tendons or the mouse tendons evaluated in the current study?

END OF COMMENTS

Dear Dr. Greenhaff,

We are grateful for the reviewers' helpful critiques and the opportunity to revise our submission number JP-RP-2024-286609.

As requested, we have responded to the individual comments of each reviewer below. The comments from each reviewer are listed in bold font, our responses are listed in italic font, and specific changes to the manuscript made to address the comments are listed below our responses.

Respectfully,

Jonathon L. Blank, PhD
Jeremy D. Eekhoff, PhD
Louis J. Soslowsky, PhD

EDITOR COMMENTS

This is an interesting and well written/executed study regarding a controversial topic in the field. Addressing the reviewer comments will strengthen the manuscript for publication.

This manuscript has been considered by a reviewing editor and two expert reviewers. The study is well powered and all believe this original research is well written. The Reviewing Editor and Reviewer 2 show enthusiasm (the latter has raised several points that the authors need to address). However, Reviewer 2 is less enthusiastic about the scientific impact of the work described and has raised a number of major concerns. These include a need for regional tendon analyses, and that all mice (not only the young) should have been included in histological analysis. It is important the authors address these concerns in a comprehensive manner when revising the manuscript, including the inclusion of additional new data if possible. Thank you for considering the Journal of Physiology to publish this work and we look forward to receiving the revised manuscript.

Authors' Response:

We thank you for your review and the summary of the reviewer comments, which are addressed in the response below.

REFeree COMMENTS

Referee #1: This manuscript examines the role of GAGs on Achilles tendon mechanical properties during aging. The premise of the manuscript is well presented in that tendon mechanical properties changes and the GAG content also changes with age so do GAGs play a role in the mechanical changes. The manuscript is well written. Several concerns with this manuscript are the lack of full analysis of the GAGs biochemically (full tendon analysis instead of regional analyses given the expected GAG difference between insertion and midsubstance; only the young mice analyzed histologically) and the limited new information generated compared to previous literature.

Comment 1.1: Line 128: If it is known that the insertion and midsubstance have different biochemical compositions, then why were they not examined separately?

Authors' Response:

We performed our biochemical assays on the entire tendon to perform a whole-tissue compositional analysis (Fig. 1C-D). All biochemical assays were performed in triplicate to be rigorous. If performed regionally, we would use a small fraction of the tissue (e.g., the 1 mm insertion region), where there would be concern of having too little tissue to perform the assay in triplicate with a GAG concentration that is detectable by our microplate assay. Thus, our regional analysis was performed using cryohistology on the 1 mm insertion and 3 mm midsubstance regions where positive toluidine blue stain corresponds to positive GAG staining (Fig. 1A-B,E). The revised manuscript clarifies that our DMMB assay was performed on the whole tendon, whereas the histologic assays provide regional information.

In Methods:

...Biochemical assays were used to determine the composition of the whole tendon. GAG concentration was measured in each Achilles tendon using a 1,9-dimethylmethylene blue (DMMB) spectrophotometric...

...Cryohistology was used to determine the regional composition of the Achilles tendon. Achilles tendons were harvested from P150 mouse hindlimbs immediately following sacrifice (n = 6/group) and treated according to the same treatment protocol...

Comment 1.2: Line 152: Why were the middle and old age tendons not examined?

Authors' Response:

We performed histology to verify that our chondroitinase digestion protocol depleted glycosaminoglycans in all regions of the tendon (i.e., insertion and midsubstance regions). We performed this analysis in P150 mice because they had a higher GAG content than P300 and P570 tendons. To compare across age, we performed the biochemical analysis in Fig. 1C-D.

Comment 1.3: Lines 224 - 228: Is the difference in the treated tendons different than the increases seen with the control samples for age?

Authors' Response:

There was no interaction between treatment and age for either insertional CSA ($p=0.747$) or midsubstance CSA ($p=0.599$). Thus, we did not detect a difference in the increases in CSA with age between control and treatment groups.

Comment 1.4: Figure 1: It seems very odd that the reduction is the exact same across all 3 ages when they older tendons have less GAGs. Please provide an explanation of this finding.

Authors' Response:

Using our digestion protocol, P150 tendon GAG content was reduced by 65.8%, P300 tendons by 62.3%, and P570 tendons by 62.3%. The similar net reduction could be because the concentration of enzyme used (0.5 U/mL cABC) can only digest a given number of GAGs under the incubation conditions in this study (18 hours at 37°C). Further, we wouldn't expect the enzyme to completely deplete the tendon of GAGs, as prior studies in cartilage (a relatively porous and GAG-enriched tissue compared to tendon) found that chondroitinase could only digest 80-90% of the GAGs without the addition of a more aggressive enzyme, such as trypsin, which we did not use due to concern of digesting collagen.

Comment 1.5: Figure 2: It seems panel D is mislabeled and should be stiffness of the Midsubstance region. Please clarify how the midsubstance stiffness remains unchanged and the midsubstance CSA significantly changes with age, but there is no effect on the elastic modulus.

Authors' Response:

The labels reported for the panels are correct. In figure 2D, we show the elastic modulus computed using the grip strain, thus reflecting the elastic modulus of the entire tissue. In 2E and 2F, we report the elastic modulus computed using the optical strains at the insertion and midsubstance regions, which reflect the modulus in those two regions respectively. Midsubstance CSA does increase with age in these tendons ($p < 0.0001$), however, this is not a 1:1 relationship with tissue stiffness or elastic modulus due to the inhomogeneity in axial strain across the length of the tendon, which is accounted for in only the regional material property measurements. We have clarified the differences in tissue vs regional properties in the text.

In Discussion:

...Specifically, there was an increase in **regional elastic** modulus in the Achilles tendon midsubstance region by a factor of 2.5-5 as compared to the insertion region.

Interestingly, the **regional elastic** modulus at the insertion increased by a factor of 1.6 when GAGs were digested from the tissue. Given that GAG content was much higher in

P150 tendons and higher at the insertion than at the midsubstance, this finding indicates that regional accumulations of GAGs may modulate **regional** tissue elasticity...

...Changes in GAG content across age do not explain age-related changes in tensile tissue mechanics of the Achilles tendon. We found that cABC treatment had no effect on **tissue-level** structural properties, viscoelastic properties, or fibril realignment in any age group...

Referee #2: This study evaluated the role of GAGs in tensile mechanics of tendons from young, middle-aged, and old adult mice. This topic has been debated for many years, with previous studies sometimes drawing contradictory conclusions about the role of GAGs in dictating tendon mechanical properties. The current study was well organized, executed with strong experimental techniques/analysis, and summarized clearly in a well-written manuscript. There are a few questions and concerns that should be addressed:

Comment 1.1: How was the incubation time selected? Does the GAG removal reach a plateau or would a longer incubation time cause further reduction in GAGs? Do other tissue components start to break down (e.g., collagen) under longer enzymatic incubation times? Also, were tendons removed from the chondroitinase ABC and rinsed at the conclusion of the 18-hour period or kept in the solution until testing?

Authors' Response:

Our digestion protocol was based on prior studies examining tendon and ligament mechanics following GAG depletion [1-2] and we chose 18 hours to ensure high efficacy of the digestion using the 0.5 U/mL concentration of cABC. We expect that the GAG removal would plateau when there are no GAGs left to digest. However, prior literature in cartilage (which is heavily enriched with GAGs compared to tendon) showed that around 80% of GAGs can be depleted using cABC, and a supplemental digestion using a tougher enzyme (trypsin) is needed to deplete nearly all of the GAGs in the tissue [3]. We could consider modifying our protocols for future studies to achieve a higher yield digestion, either by digesting for longer, using a higher concentration (e.g., 1U/mL cABC), or using a supplemental trypsin digestion.

We were concerned about collagens breaking down under the long 18-hour digestion time, which is why we performed the OHP assay. Importantly, we found no difference in collagen content after 18 hours of incubation. However, it is possible that longer digestion times could promote collagen degradation, and this would need to be evaluated in any future study. Finally, we did rinse tendons in 1X PBS following digestion prior to histology, biochemistry, or biomechanics.

In Methods:

...Immediately following fine dissection, each tendon-calcaneus unit was immersed in 0.33 mL of a Tris buffer supplemented with protease inhibitors (pH = 8.0) for one hour at 37°C and under gentle agitation [27]. Following the initial buffer incubation, tendons were transferred to 0.33 mL of the same buffer solution (control), or the buffer solution supplemented with 0.5 U/mL chondroitinase ABC (cABC, Sigma #C3667) (experimental treatment), which selectively cleaves sulfated GAGs from their proteoglycan core proteins. Tendons were incubated in the secondary buffer for 18 hours at 37°C under gentle agitation. **All tendons were rinsed in 1X PBS following incubation. cABC concentration and incubation steps were chosen based on prior studies examining tendon and ligament mechanics following GAG depletion (Lujan et al., 2009; Rigozzi et al., 2009), and an 18-hour incubation time was chosen to ensure high efficacy of the digestion using the 0.5 U/mL cABC concentration.** All

biochemical, biomechanical, and histological procedures took place after this initial control or experimental treatment...

...Collagen content was measured **following digestion** using a hydroxyproline (OHP) assay **to ensure that collagen was not degraded by incubation in the enzyme or buffer** (n = 4-5/group) [30]. The tendon was hydrolyzed with 12N HCl in a sealed container at 110°C for 48 hours...

References

- [1] S. Rigozzi, R. Müller, and J. G. Snedeker, "Local strain measurement reveals a varied regional dependence of tensile tendon mechanics on glycosaminoglycan content," *J. Biomech.*, vol. 42, no. 10, pp. 1547–1552, 2009, doi: 10.1016/j.jbiomech.2009.03.031.
- [2] T. J. Lujan, C. J. Underwood, N. T. Jacobs, and J. A. Weiss, "Contribution of glycosaminoglycans to viscoelastic tensile behavior of human ligament," *J. Appl. Physiol.*, vol. 106, no. 2, pp. 423–431, 2009, doi: 10.1152/jappphysiol.90748.2008.
- [3] M. B. Schmidt, V. C. Mow, L. E. Chun, and D. R. Eyre, "Effects of proteoglycan extraction on the tensile behavior of articular cartilage," *J. Orthop. Res.*, vol. 8, no. 3, pp. 353–363, 1990, doi: 10.1002/jor.1100080307.

Comment 1.2: The histological images shown for P150 tendons in Figure 1A and 1B are beautiful and insightful. Why was histological analysis not performed for the P300 and P570 mice? It would be very helpful to include images from these two other ages so that similarities and differences by age can be observed. In addition, the positive stain fraction values could be computed/reported for the older mice too.

Authors' Response:

As noted in response to Reviewer 1, we performed histology to verify that our chondroitinase digestion protocol depleted glycosaminoglycans in all regions of the tendon (i.e., insertion and midsubstance regions). We performed this analysis in P150 mice because they had a higher GAG content than P300 and P570 tendons. To compare across age, we performed the biochemical analysis in Fig. 1C-D.

Comment 1.3: What is the hypothesized mechanism for why insertion elastic modulus increased in P150 tendons following GAG depletion?

Authors' Response:

We hypothesize that GAGs mediate frictional forces between adjacent fibrils by drawing water into the tendon ECM. Thus, when fewer GAGs are present, there are higher forces within tendon fibers at an equivalent strain due to less overall sliding between adjacent fibrils, as postulated by Rigozzi et al. [1]. Ongoing experiments are working to address this hypothesis in young and aged mice.

References

[1] S. Rigozzi, R. Müller, A. Stemmer, and J. G. Snedeker, "Tendon glycosaminoglycan proteoglycan sidechains promote collagen fibril sliding-AFM observations at the nanoscale," *J. Biomech.*, vol. 46, no. 4, pp. 813–818, 2013, doi: 10.1016/j.jbiomech.2012.11.017.

Comment 1.4: Figure 4 only shows data for P150 (a-b) but the caption suggests that P300 (c-d) and P570 (e-f) were meant to be included. Where are these additional data?

Authors' Response:

We regret this error in the preparation of this manuscript. We did include the data for P300 and P570 tendons in the supplementary info but will now include them in the main body of the manuscript.

In Fig. 4:

Figure 4: Fibril realignment patterns across age and treatment. We did not observe changes in fibril realignment patterns in either the insertion (**a,c,e**) or midsubstance (**b,d,f**) regions in P150 (**a-b**), P300 (**c-d**) or P570 (**e-f**) Achilles tendons. Data shown as mean \pm standard deviation.

Comment 1.5: If the collagen content (i.e., OHP) and GAG content both decrease in the P570 tendons, what compositional constituents of these older tendons are increased? In other words, what are these tendons becoming with increased age?

Authors' Response:

We believe that heterotopic ossification (HO, fibrocartilage and bone formation) found commonly in murine tendons [1] is a driver of these compositional changes. Ossification deposits were noted in fine dissections in both P300 and P570 mice, with generally more HO occurring in the P570 mice. Further, normalization to dry weight favors these changes as HO is denser than tendon and thus would increase the dry weight of the tendons more than natural aging.

References

[1] G. Dai *et al.*, "Higher BMP Expression in Tendon Stem/Progenitor Cells Contributes to the Increased Heterotopic Ossification in Achilles Tendon With Aging," *Front. Cell Dev. Biol.*, vol. 8, no. September, pp. 1–14, 2020, doi: 10.3389/fcell.2020.570605.

Comment 1.6: Can the authors discuss how GAG composition in mouse Achilles tendons compares to human Achilles tendons? Are the relative percentages of GAG similar or quite different across the two species? And what is known about GAG degradation with age in human Achilles tendons? What implications does this have for interpreting the current results in the context of human tissues?

Authors' Response:

*To our knowledge, there are no prior studies quantifying GAG content in the native human Achilles tendon. Riley *et al.* quantified GAG content in the human rotator cuff and biceps tendon, finding concentrations of 12.3 and 1.2 $\mu\text{g}/\text{mg}$, respectively [1]. Our measured GAG content in the healthy mouse Achilles tendon (1.95-3.31 $\mu\text{g}/\text{mg}$) is between the range of these concentrations, though there is considerable variability in GAG content between tendon types. The same prior study concluded that GAG content decreased with age in tendon, although the Achilles tendon was not their focus. Thus, our biochemical findings are consistent with those across age in this prior study.*

Our finding was that GAGs may influence mechanics at the Achilles tendon insertion in the absence of other changes. This may apply to the human Achilles, however, there are several considerable differences between the mouse and human Achilles, including size, loading magnitudes relative to bodyweight, and also changes to the tendon

hierarchy, where mouse Achilles tendons lack fascicles (i.e., bundles of fibers). Further, similar to the findings of Riley et al. in the supraspinatus, there is likely considerable variability in GAG content between individuals due to tendinopathy, rupture, and metabolic disorders, among other conditions.

We have also revised Figure 1 to include the plots of non-normalized results (which were originally added as supplemental information), as this has been a major discussion point in this review and may aid readers in the interpretation of our results.

In Discussion:

...Regional optical tracking of Achilles tendons showed a dramatic difference in modulus between the insertion and midsubstance regions of the tissue. Specifically, there was an increase in modulus in the Achilles tendon midsubstance region by a factor of 2.5-5 as compared to the insertion region. Interestingly, the modulus at the insertion increased by a factor of 1.6 when GAGs were digested from the tissue. Given that GAG content was much higher in P150 tendons and higher at the insertion than at the midsubstance, this finding indicates that regional accumulations of GAGs may modulate tissue elasticity. Interestingly, a prior study examining regional tissue mechanics within the Achilles tendon midsubstance following GAG depletion found a decreased modulus in the distal portion of the Achilles tendon midsubstance in young tendons [26], which contrasts our finding of an increased modulus at the insertion following digestion. However, the distal region of tendon in the prior study was still proximal to the tendon insertion, whereas our regional measurements include the tendon insertion into bone and likely captures regions of the tendon near the insertion that have enriched GAG content. ***Finally, it is unknown whether GAGs would influence regional properties in humans, as there are differences between the mouse and human Achilles, including size, loading magnitudes relative to bodyweight, and changes to the tendon hierarchy, where mouse Achilles tendons lack fascicles (i.e., bundles of fibers)...***

...GAG content is known to decrease across age in many tissues, such as in cartilage [37] and meniscus [38]. We show that GAG content relative to the dry weight of the ECM in the free Achilles tendon decreases from 0.33% in P150 tendons to 0.16% in P570 tendons, and similarly that collagen content in the free tendon decreased from nearly 100% collagen in P150 tendons to 66% collagen in P570 tendons (Supplemental Fig. 1). While decreases in GAG and collagen content have been found with age in adult tendon [19], [39], the differences reported in this study are likely exaggerated due to increased heterotopic ossification in the rodent Achilles tendon across age [36]. Thus, decreases in GAG and OHP content with age in the ***mouse*** Achilles tendon are likely to be more modest than reported in this study. ***To our knowledge, there are no prior studies quantifying GAG content in the native human Achilles tendon. Riley et al. quantified GAG content in the human rotator cuff and biceps tendon, finding concentrations of 12.3 and 1.2 µg/mg, respectfully (Riley et al., 1994). Our measured GAG content in the healthy mouse Achilles tendon (3.31 µg/mg at P150, 2.40 µg/mg at P300, and 1.95 µg/mg at P570) is between the range of these concentrations, yet there is considerable variability in GAG content between***

tissue type. Changes in GAG content with age may also be loading environment-dependent, as a prior study in murine flexor digitorum longus tendons found a similar decrease in GAG content with age when normalized to the dry weight [40], and increased GAG synthesis in younger tendons when stress-deprived [41]. Investigating whether the Achilles tendon can modulate GAG production to accommodate changes in loading environment may elucidate age-related pathogenic changes to tendon associated with altered GAG content, such as in insertional Achilles tendinopathy [42], [43]...

In Figure 1:

Figure 1: GAG content across age and region following treatment in the Achilles tendons. There were more GAGs in control tendons (A) than cABC treated tendons (B)

(P150 tendons shown, intensity range 50-200, scale bar = 100 μ m). **(C-D) GAG and collagen content decreased similarly throughout age ($p = 0.0080$ and $p < 0.0001$, respectively).** **(E-F)** cABC treatment removed 64% of GAGs on average from the entire Achilles tendon ($p < 0.0001$) without any changes to collagen content ($p = 0.808$). **(G)** there were fewer GAGs in the Achilles midsubstance **(i)** than at the insertion **(ii)** (main region effect $p = 0.0043$), yet GAGs were digested similarly in the insertion and midsubstance regions (main digestion effect $p < 0.0001$). Data presented as mean \pm standard deviation (**** $p < 0.0001$) (# = main digestion effect, & = main region effect).

References

[1] G. P. Riley, R. L. Harrall, C. R. Constant, M. D. Chard, T. E. Cawston, and B. L. Hazleman, "Glycosaminoglycans of human rotator cuff tendons: Changes with age and in chronic rotator cuff tendinitis," *Ann. Rheum. Dis.*, vol. 53, no. 6, pp. 367–376, 1994, doi: 10.1136/ard.53.6.367.

Comment 1.7: The assertions that GAGs (1) "likely regulate the elastic modulus" in the insertion region and (2) "modulate regional properties at the Achilles tendon insertion in young tendons" seem too strong given that cABC effects were only (and just barely) significant in young tendons ($p=0.0495$). Many of the data points for the ChABC-treated P150 tendons fell within the range of untreated tendons (Figure 2E), and it appears that larger modulus values from four tendons (~600-900 MPa) are likely driving the statistical difference between untreated and treated groups. That's not to say these aren't groups aren't different...but just that conclusions based on these differences should be appropriately tempered.

Authors' Response:

We agree and have tempered these conclusions accordingly. Specifically, we now state that GAGs "may influence" mechanical properties at the insertion.

In Discussion:

...Given that GAG content was much higher in P150 tendons and higher at the insertion than at the midsubstance, this finding indicates that regional accumulations of GAGs may **influence** tissue elasticity...

...This study helps to elucidate the role of GAGs in tensile tendon mechanics throughout aging, finding that GAGs **may influence** regional properties at the Achilles tendon insertion in young tendons. Further, age-related changes in GAG content are not responsible for changes to structural properties, viscoelasticity or fibril realignment patterns...

Comment 1.8: In the discussion, is the sentence that reads: "While decreases in GAG and collagen content have been found with age in adult tendon [19],[39] ..." referring to adult human tendon? Also, does the sentence: "Thus, decreases in GAG and OHP content with age in the Achilles tendon are likely to be more modest than reported in this

study" refer to human tendons or the mouse tendons evaluated in the current study?

Authors' Response:

For the first discussion sentence, the decreases in GAG content were found in human tendon, and decreases in collagen content were found in canine tendon. Concerning the latter sentence, we expect that decreases in GAG content with age would be most applicable to murine tendon in this context. We have adjusted the discussion accordingly.

In Discussion:

...Thus, decreases in GAG and OHP content with age in the **mouse** Achilles tendon are likely to be more modest than reported in this study...

Dear Dr Soslowsky,

Re: JP-RP-2025-286609R1 "Glycosaminoglycans influence regional mechanics in young but not old Achilles tendons" by Jonathon L Blank, Jeremy Eekhoff, and Louis Soslowsky

Thank you for submitting your revised manuscript to The Journal of Physiology. It has been assessed by a Reviewing Editor and by 2 expert referees and we are pleased to tell you that it is acceptable for publication following satisfactory revision.

REVISION CHECKLIST:

We look forward to receiving your revised submission.

Yours sincerely,

Paul Greenhaff
Senior Editor
The Journal of Physiology

REQUIRED ITEMS FOR REVISION

- You must start the Methods section with a paragraph headed Ethical approval (https://jp.msubmit.net/cgi-bin/main.plex?form_type=display_requirements#methods).

Research must comply with The Journal's policies regarding animal experiments (<https://physoc.onlinelibrary.wiley.com/hub/animal-experiments>) and adherence to these policies must be stated in the manuscript.

Authors should confirm in their Methods section that their experiments were carried out according to the guidelines laid down by their institution's animal welfare committee, including an ethics approval reference number. The Methods section must contain a statement about access to food, water and housing, details of the anaesthetic regime: anaesthetic used, dose and route of administration, and method of killing the experimental animals.

- Papers must comply with the Statistics Policy: https://jp.msubmit.net/cgi-bin/main.plex?form_type=display_requirements#statistics.

In summary:

- If $n \leq 30$, all data points must be plotted in the figure in a way that reveals their range and distribution. A bar graph with data points overlaid, a box and whisker plot or a violin plot (preferably with data points included) are acceptable formats.
- If $n > 30$, then the entire raw dataset must be made available either as supporting information, or hosted on a not-for-profit repository, e.g. FigShare, with access details provided in the manuscript.
- 'n' clearly defined (e.g. x cells from y slices in z animals) in the Methods. Authors should be mindful of pseudoreplication.
- All relevant 'n' values must be clearly stated in the main text, figures and tables.
- The most appropriate summary statistic (e.g. mean or median and standard deviation) must be used. Standard Error of the Mean (SEM) alone is not permitted.
- Exact p values must be stated. Authors must not use 'greater than' or 'less than'. Exact p values must be stated to three significant figures even when 'no statistical significance' is claimed.

- Please include an Abstract Figure file, as well as the Figure Legend text within the main article file. The Abstract Figure is a piece of artwork designed to give readers an immediate understanding of the research and should summarise the main conclusions. If possible, the image should be easily 'readable' from left to right or top to bottom. It should show the physiological relevance of the manuscript so readers can assess the importance and content of its findings. Abstract Figures should not merely recapitulate other figures in the manuscript. Please try to keep the diagram as simple as possible and without superfluous information that may distract from the main conclusion(s). Abstract Figures must be provided by authors no later than the revised manuscript stage and should be uploaded as a separate file during online submission labelled as File Type 'Abstract Figure'. Please also ensure that you include the figure legend in the main article file. All Abstract Figures should be created using BioRender. Authors should use The Journal's premium BioRender account to export high-resolution

images. Details on how to use and access the premium account are included as part of this email.

EDITOR COMMENTS

Reviewing Editor:

The manuscript has been re-evaluated by the original reviewers, both of whom acknowledge that several concerns have been addressed, resulting in significant improvements. However, some issues remain, including the need for histological analysis from the older age group of mice and GAG digestion.

The authors are encouraged to address these remaining concerns.

Senior Editor:

Thank you for the revised manuscript which has been considered by the same reviewers that considered the original submission. Both feel that the authors did an adequate job at responding to the comments raised in review, but there are divergent opinions regarding the scientific impact of the work. The authors are encouraged to fully address the concerns of Reviewer 2, including the further analytical work requested, to assist in progression of the manuscript to publication.

REFEREE COMMENTS

Referee #1:

Line 172: These biochemistry experiments should have been done regionally. Doing these experiments over the whole tendon masks the regional changes in the insertion site. This is a quantitative measure of how much GAG was lost due to the treatment. The standards were set from 0 - 32 µg/ml. This should have allowed for the smaller regional samples.

Lines 201 - 202: If that is truly the case, then why not perform it at all ages.

Lines 264 - 270: Given that the GAG levels decreased with time, but the level of enzyme was consistent, it is still unclear to me how the values are the same percentage.

Referee #2:

The authors addressed several of my previous questions and critiques. However, a few concerns remain:

- I still believe that the authors should add histological analysis of tendons from older mice. In addition to making this a more complete and consistent study across all groups, this approach would elucidate the morphology (and changes following cABC digestion) of older tendons. In addition, these analyses would allow examination of regional changes in the tendons (and not just whole tissue biochemistry).
- The authors reported that they "expect that the GAG removal would plateau when there are no GAGs left to digest." However, the authors noted that previous studies have shown digestion of only 80-90% in cartilage (rather than 100%). In the current study, they report ~62-65% digestion of GAG. Thus, it appears that under cABC digestion a plateau would be reached before all GAGs are digested. Thus, the question remains whether the digestion protocol used in this study was at a plateau point or whether more GAGs could have been digested under longer incubation.
- In response to a previous question, the authors hypothesized that the elastic modulus at the insertion increased in P150 tendons following GAG depletion because of higher frictional forces after GAG removal. However, several previous studies have shown no difference (or decreased) tensile mechanics following GAG depletion (e.g., Rigozzi 2009). If the increased frictional force hypothesis were correct, wouldn't increased modulus values have been observed in these other studies following GAG removal? Even if the GAG amount was less in other regions, wouldn't the effect still be the same if the

authors' hypothesis is correct?

END OF COMMENTS

Dear Dr. Greenhaff,

We are grateful for the reviewers' helpful critiques and the opportunity to revise our submission number JP-RP-2024-286609.

As requested, we have responded to the individual comments of each reviewer below. The comments from each reviewer are listed in bold font, our responses are listed in italic font, and specific changes to the manuscript made to address the comments are listed below our responses.

Respectfully,

Jonathon L. Blank, PhD
Jeremy D. Eekhoff, PhD
Louis J. Soslowsky, PhD

EDITOR COMMENTS

Reviewing Editor:

The manuscript has been re-evaluated by the original reviewers, both of whom acknowledge that several concerns have been addressed, resulting in significant improvements. However, some issues remain, including the need for histological analysis from the older age group of mice and GAG digestion.

The authors are encouraged to address these remaining concerns.

Senior Editor:

Thank you for the revised manuscript which has been considered by the same reviewers that considered the original submission. Both feel that the authors did an adequate job at responding to the comments raised in review, but there are divergent opinions regarding the scientific impact of the work. The authors are encouraged to fully address the concerns of Reviewer 2, including the further analytical work requested, to assist in progression of the manuscript to publication.

Authors' Response:

We thank you for your review and the summary of the reviewer comments, which are addressed in the response below.

REFEREE COMMENTS

Referee #1:

Comment 1.1: Line 172: These biochemistry experiments should have been done regionally. Doing these experiments over the whole tendon masks the regional changes in the insertion site. This is a quantitative measure of how much GAG was lost due to the treatment. The standards were set from 0 - 32 µg/ml. This should have allowed for the smaller regional samples.

Authors' Response:

Unfortunately, the small size of the insertion relative to the whole tendon makes the normalization procedure of the biochemical analysis challenging. A pilot experiment revealed dry weights as low as 0.04 mg in the young Achilles tendon insertion region (0-1 mm from the calcaneus), which is close to the resolution of our standard microscale (0.01 mg, resulting in 25% measurement uncertainty). Future studies will explore normalizing to protein levels within each region. This has been acknowledged in the discussion.

In Discussion:

...Third, a higher efficacy digestion protocol (i.e., greater than 64%) such as incubation for a longer time, using a secondary digestion (Schmidt et al., 1990), or using an increased cABC concentration (i.e., 1 U/mL) may have increased the mechanical changes observed with GAG removal. Finally, performing region-specific biochemical assays would have provided additional explanatory power for regional differences in the mechanical changes following GAG removal...

Comment 1.2: Lines 201 - 202: If that is truly the case, then why not perform it at all ages.

Authors' Response:

We have added histological analysis of the P300 and P570 age groups as requested, in addition to new data for the P150 mice. Prior histology data for the P150 mice was collected on fresh tendon (i.e., not stored). Given that all the mice (aside from the original histology) were frozen at -20C prior to performing the experiments, we have recollected all the histology on frozen P150, P300, and P570 mice to better represent the samples used for biochemistry and mechanics.

In Methods:

...Following euthanasia by carbon dioxide inhalation, mice were weighed (P150: 30.6 ± 3.0 g, P300: 35.7 ± 3.6 g, P570: 38.4 ± 6.0 g). All animals were frozen at -20°C until further use. Once thawed, Achilles tendons were harvested from mouse hindlimbs and cleaned from surrounding tissues with the calcaneus intact...

...Cryohistology was used to determine the regional composition of the Achilles tendon. **Achilles tendons were harvested from frozen P150, P300, and P570 mouse hindlimbs (n = 3/group)** and treated according to the same treatment protocol. Preparation for imaging was designed based on a prior cryohistology protocol (Dyment et al., 2016). The Achilles tendon was fixed at 90° to the calcaneus in a cassette and immersed in 10% neutral-buffered formalin for 4 hours. Samples were then soaked overnight in a 30% sucrose solution and embedded in optimal cutting temperature compound. Sections were cut sagittally using cryotape at an 8 µm thickness, glued to glass slides using 1% chitosan in acetic acid, and dried overnight. Slides were then rehydrated in distilled water, stained in 0.025% toluidine blue for 3 minutes and rinsed, coverslipped using 30% fructose mounting medium, and imaged at 20X in a slide scanner (Zeiss Axioscan). **Following color deconvolution**, positive purple stain fraction in the insertion (0-1 mm from insertion tidemark) and midsubstance (1-3 mm from insertion tidemark) was used to quantify the presence of GAGs. Comparisons were made between contralateral limbs receiving either treatment.

In Results:

... Histology showed that cABC treatment depleted GAGs similarly from the Achilles tendon midsubstance ($p = 0.0012$, Fig. 1A-B) and insertion ($p < 0.0001$, Fig 1C-D) across age. Additionally, there was heterotopic ossification (HO) in the Achilles tendon midsubstance of the P300 and P570 mice, as visualized by undigested areas of GAG concentration (red arrows, Fig. 1A). Generally, GAG content and collagen content decreased with age (Fig. 1E-F). When normalized to the average value for controls within each age group due to the presence of HO, our digestion protocol showed a 64% depletion of GAGs in the Achilles tendon across all ages ($p \leq 0.0001$) without changes to collagen content ($p = 0.808$) (Fig. 1G-H)...

In Discussion:

...While decreases in GAG and collagen content have been found with age in adult tendon (Haut et al., 1992; Riley et al., 1994), the differences reported in this study are likely exaggerated due to increased heterotopic ossification in the rodent Achilles tendon across age (Fig. 1A) (Dai et al., 2020). **Thus, decreases in GAG and OHP content with age in the mouse Achilles tendon are likely to be more modest than reported in this study. Concerning digestion, we found that decreases in positive GAG staining with digestion were similar across age in the Achilles tendon insertion and midsubstance. We also observed HO in the aging tendon midsubstance, as visualized by undigested concentrations of GAGs in bony regions of the tendon (Fig. 1A), which may have contributed to less overall GAGs being digested with increased age (i.e., 2.18 µg GAG per mg tissue dry weight digested at P150 vs 1.21 µg GAG per mg tissue dry weight digested at P570). Interestingly, despite these regions of ossification in the aging tendon midsubstance, we did not detect any differences in overall tendon stiffness, elastic modulus, or midsubstance elastic modulus across age, indicating that these HO deposits did not alter tissue level tensile mechanics.**

In Figures:

Figure 1: GAG content across age and region following treatment in the Achilles tendons. There were more GAGs in control tendons than cABC treated tendons in both the midsubstance (A-B) and insertion regions (C-D) (intensity range 0-200, scale bar = 100 µm). In the tendon midsubstance, there was notable amounts of heterotopic ossification with increased age, as shown by lack of consistent stain and mottled GAG concentrations (see red arrows). GAGs were depleted similarly across age in the midsubstance (A) and insertion (B) at P150 ($p = 0.0012$ and $p < 0.0001$, respectively). (E-F) GAG and collagen content decreased similarly throughout age ($p = 0.0080$ and $p < 0.0001$, respectively). (G-H) cABC treatment removed 64% of GAGs on average from the entire Achilles tendon ($p < 0.0001$) without any changes to collagen content ($p = 0.808$). Data presented as mean \pm standard deviation (* $p < 0.05$, ** $p < 0.01$, *** $p < 0.001$, **** $p < 0.0001$) (main effects, # $p < 0.01$, & $p < 0.0001$).

Comment 1.3: Given that the GAG levels decreased with time, but the level of enzyme was consistent, it is still unclear to me how the values are the same percentage.

Authors' Response:

There were relatively fewer GAGs digested in the aged groups. Chondroitinase digestion removed 2.18 µg/mg, 1.76 µg/mg, and 1.21 µg/mg dry weight on average from P150, P300, and P570 tendons, respectively. Percentagewise, these digestions work out to be about the same across age (62-65%). A likely explanation for fewer depleted

GAGs in the aged groups is heterotopic ossification (HO). Not only does HO increase the dry weight taken before the proteinase homogenization (thus decreasing GAGs/dry weight), but HO contains a considerable amount of GAGs that remained undigested (see dark purple staining in ages samples treated with chondroitinase, Fig. 1).

In Discussion:

...While decreases in GAG and collagen content have been found with age in adult tendon (Haut et al., 1992; Riley et al., 1994), the differences reported in this study are likely exaggerated due to increased heterotopic ossification in the rodent Achilles tendon across age (**Fig. 1A**) (Dai et al., 2020). **Thus, decreases in GAG and OHP content with age in the mouse Achilles tendon are likely to be more modest than reported in this study. Concerning digestion, we found that decreases in positive GAG staining with digestion were similar across age in the Achilles tendon insertion and midsubstance. We also observed HO in the aging tendon midsubstance, as visualized by undigested concentrations of GAGs in bony regions of the tendon (Fig. 1A), which may have contributed to less overall GAGs being digested with increased age (i.e., 2.18 μg GAG per mg tissue dry weight digested at P150 vs 1.21 μg GAG per mg tissue dry weight digested at P570)...**

Referee #2:

Comment 1.1: I still believe that the authors should add histological analysis of tendons from older mice. In addition to making this a more complete and consistent study across all groups, this approach would elucidate the morphology (and changes following cABC digestion) of older tendons. In addition, these analyses would allow examination of regional changes in the tendons (and not just whole tissue biochemistry).

Authors' Response:

As mentioned in the response to Reviewer #1, we have added histological analysis of the P300 and P570 age groups as requested, in addition to new data for the P150 mice. Prior data for the P150 mice was collected on fresh tendon (i.e., not stored). Given that all the mice (aside from the original histology) were frozen, we have recollected all the histology on frozen P150, P300, and P570 mice to better represent the samples used for biochemistry and mechanics. The changes to the manuscript can be seen in the response to Reviewer #1.

Comment 1.2: The authors reported that they "expect that the GAG removal would plateau when there are no GAGs left to digest." However, the authors noted that previous studies have shown digestion of only 80-90% in cartilage (rather than 100%). In the current study, they report ~62-65% digestion of GAG. Thus, it appears that under cABC digestion a plateau would be reached before all GAGs are digested. Thus, the question remains whether the digestion protocol used in this study was at a plateau point or whether more GAGs could have been digested under longer incubation.

Authors' Response:

Importantly, the digestion could also change with tissue type. Prior studies in isolated medial collateral ligaments with no bony attachments found a GAG digestion as high as 88% using 1 U/mL for only 6 hours [1], while a study in rat tail tendon fascicles found a digestion of 93% following 18 hours in 0.3 U/mL cABC [2]. A prior study in the isolated soleus subtendon of the Achilles tendon found a 60% reduction in only 0.15 U/mL cABC for 12 hours [3], however when digesting the entire tendon (including the tendon insertion to bone) this digestion only digested 47-54% of GAGs using the same protocol [4]. Thus, there seems to be some variation depending on tissue type and region, even when the protocol is kept the same.

We chose the 0.5 U/mL concentration and 18-hour incubation time based on what was shown to be effective in tendon in prior literature. We agree that longer incubation time, a secondary digestion, or higher cABC concentration could all potentially digest more GAGs from the tendon. This has been acknowledged in the discussion.

In Discussion:

... Third, a higher efficacy digestion protocol (i.e., greater than 64%) such as incubation for a longer time, using a secondary digestion (Schmidt et al., 1990), or using an increased cABC concentration (i.e., 1 U/mL) may have increased the

mechanical changes observed with GAG removal. Finally, performing region-specific biochemical assays would have provided additional explanatory power for regional differences in the mechanical changes following GAG removal...

References:

- [1] Henninger, H. B., Underwood, C. J., Ateshian, G. A., & Weiss, J. A. (2010). Effect of Sulfated Glycosaminoglycan Digestion on the Transverse Permeability of Medial Collateral Ligament. *Journal of Biomechanics*, 43(13), 2567–2573.
- [2] Fessel, G., & Snedeker, J. G. (2009). Evidence against proteoglycan mediated collagen fibril load transmission and dynamic viscoelasticity in tendon. *Matrix Biology*, 28(8), 503–510.
- [3] Rigozzi, S., Müller, R., Stemmer, A., & Snedeker, J. G. (2013). Tendon glycosaminoglycan proteoglycan sidechains promote collagen fibril sliding-AFM observations at the nanoscale. *Journal of Biomechanics*, 46(4), 813–818.
- [4] Rigozzi, S., Müller, R., & Snedeker, J. G. (2009). Local strain measurement reveals a varied regional dependence of tensile tendon mechanics on glycosaminoglycan content. *Journal of Biomechanics*, 42(10), 1547–1552.

Comment 1.3: In response to a previous question, the authors hypothesized that the elastic modulus at the insertion increased in P150 tendons following GAG depletion because of higher frictional forces after GAG removal. However, several previous studies have shown no difference (or decreased) tensile mechanics following GAG depletion (e.g., Rigozzi 2009). If the increased frictional force hypothesis were correct, wouldn't increased modulus values have been observed in these other studies following GAG removal? Even if the GAG amount was less in other regions, wouldn't the effect still be the same if the authors' hypothesis is correct?

Authors' Response:

The effect of GAGs on tensile mechanics in the tendon midsubstance is minimal, and many prior studies examined tail tendon fascicles [1] or isolated ligament midsubstance [2], which would have low GAG content as compared to the Achilles tendon insertion [3]. Given this, the hypothesis is more specific to the insertion or areas where there are accumulations of GAGs, such as in an injured tendon. It is worth noting that this concept has been explored in fibrocartilage, where areas with enriched GAG deposition exhibit recovery of ECM-tissue strain ratio (i.e., increased load transfer between fibrils) following chondroitinase depletion [4]. We have included this in the revised discussion.

In Discussion:

... Interestingly, the regional elastic modulus at the insertion increased by a factor of 1.6 when GAGs were digested from the tissue. ***Prior studies in the tendon and ligament midsubstance have found negligible impact of GAG digestion on mechanics (Fessel & Snedeker, 2009; Lujan et al., 2007; Lujan et al., 2009). However, GAG content is generally higher in the tendon insertion than the tendon midsubstance (Waggett et al., 1998), where there is increased compression, impingement, and cartilaginous features as the tendon transitions to bone (Chimenti et al., 2016;***

Maffulli et al., 2006). The higher GAG content in this fibrocartilage-rich region of the Achilles tendon may modulate regional tissue elasticity, as has been found in isolated fibrocartilage (Han et al., 2016)...

References:

- [1] Henninger, H. B., Underwood, C. J., Ateshian, G. A., & Weiss, J. A. (2010). Effect of Sulfated Glycosaminoglycan Digestion on the Transverse Permeability of Medial Collateral Ligament. *Journal of Biomechanics*, 43(13), 2567–2573.
- [2] Fessel, G., & Snedeker, J. G. (2009). Evidence against proteoglycan mediated collagen fibril load transmission and dynamic viscoelasticity in tendon. *Matrix Biology*, 28(8), 503–510.
- [3] Waggett, A. D., Ralphs, J. R., Kwan, A. P. L., Woodnutt, D., & Benjamin, M. (1998). Characterization of collagens and proteoglycans at the insertion of the human Achilles tendon. *Matrix Biology*, 16(8), 457–470.
- [4] Han, W. M., Heo, S.-J., Driscoll, T. P., Delucca, J. F., McLeod, C. M., Smith, L. J., Duncan, R. L., Mauck, R. L., Elliott, D. M. (2016). Microstructural heterogeneity directs micromechanics and mechanobiology in native and engineered fibrocartilage. *Nature Materials*, 15, 477-484.

Dear Dr Blank,

Re: JP-RP-2025-286609R2 "Glycosaminoglycans influence regional mechanics in young but not old Achilles tendons" by Jonathon L Blank, Jeremy Eekhoff, and Louis Soslowsky

We are pleased to tell you that your paper has been accepted for publication in The Journal of Physiology.

Yours sincerely,

Paul Greenhaff
Senior Editor
The Journal of Physiology

If you would like to receive our 'Research Roundup', a monthly newsletter highlighting the cutting-edge research published in The Physiological Society's family of journals (The Journal of Physiology, Experimental Physiology, Physiological Reports, The Journal of Nutritional Physiology and The Journal of Precision Medicine: Health and Disease), please click this link, fill in your name and email address and select 'Research Roundup':
<https://www.physoc.org/journals-and-media/membernews>

- **TRANSPARENT PEER REVIEW POLICY:** To improve the transparency of its peer review process, The Journal of Physiology publishes online as supporting information the peer review history of all articles accepted for publication. Readers will have access to decision letters, including Editors' comments and referee reports, for each version of the manuscript as well as any author responses to peer review comments. Referees can decide whether or not they wish to be named on the peer review history document.
- You can help your research get the attention it deserves! Check out Wiley's free Promotion Guide for best-practice recommendations for promoting your work at: www.wileyauthors.com/eeo/guide. You can learn more about Wiley Editing Services which offers professional video, design, and writing services to create shareable video abstracts, infographics, conference posters, lay summaries, and research news stories for your research at: www.wileyauthors.com/eeo/promotion.
- **IMPORTANT NOTICE ABOUT OPEN ACCESS:** To assist authors whose funding agencies mandate public access to published research findings sooner than 12 months after publication, The Journal of Physiology allows authors to pay an Open Access (OA) fee to have their papers made freely available immediately on publication.

EDITOR COMMENTS

Reviewing Editor:

Ethics Concerns:
None

The article is acceptable for publication

Senior Editor:

Thank you for the re-revised manuscript and for addressing the points raised by Reviewer 2. The manuscript is now acceptable for publication. Congratulations and thank you for considering The Journal of Physiology to publish your research.

REFEREE COMMENTS

Referee #2:

The authors addressed my concerns and comments.